# Early Childhood Caries and Its Associated Factors among 9- to 18-Month Old Exclusively Breastfed Children in Thailand: A Cross-Sectional Study

**DOI:** 10.3390/ijerph17093194

**Published:** 2020-05-04

**Authors:** Pichet Chanpum, Duangporn Duangthip, Chutima Trairatvorakul, Siriporn Songsiripradubboon

**Affiliations:** 1Nongsung Hospital, Mukdahan 49160, Thailand; juppdt@gmail.com; 2Faculty of Dentistry, The University of Hong Kong, Hong Kong SAR, China; dduang@hku.hk; 3Department of Pediatric Dentistry, Faculty of Dentistry, Chulalongkorn University, Bangkok 10330, Thailand; ctrairat@gmail.com

**Keywords:** dental caries, early childhood caries, breastfeeding, child, oral health

## Abstract

*Objective*: The objective of this study was to investigate the early childhood caries (ECC) status and its risk factors in 9- to 18-month-old exclusively breastfed children in Thailand. *Methods:* Generally healthy 9- to 18-month-old children who had been exclusively breastfed were recruited. Information on children’s oral hygiene practices and breastfeeding behaviors was collected through parental interviews using a questionnaire. Children’s oral health status was assessed following the WHO caries diagnostic criteria, modified to record the noncavitated lesions. Multivariate logistic regression analysis was adopted to investigate its association with feeding and oral hygiene practices. *Results*: In total, 513 mother and child dyads (47% boys) were recruited. The prevalence of ECC was 42.5%. The mean (SD) d_1_mft and d_1_mfs scores (d_1_ included noncavitated and cavitated carious teeth/tooth surfaces) were 1.1 (1.4) and 1.3 (2.0), respectively. Multivariate logistic regression analysis revealed that older children with higher plaque scores (OR = 75.60; 95% CI: 40.19–142.20) who were breastfed to sleep (OR = 2.85; 95% CI: 1.48–5.49) and never had their teeth cleaned (OR = 8.51; 95% CI: 1.53–47.14), had a significantly higher chance of having ECC (*p* < 0.05). *Conclusion:* Prevalence of ECC is high among exclusively breastfed children aged 9–18 months in Thailand. ECC prevalence is significantly associated with the age of children, the level of dental plaque, breastfeeding to sleep, and oral cleaning. Among all factors, the level of dental plaque is the most significant factor associated with ECC among breastfed children.

## 1. Introduction

Early childhood caries (ECC) is a global oral health concern [1]. It remains prevalent in many countries and highly prevalent in developing countries worldwide [2]. In Southeast Asia, a systematic review showed that ECC prevalence was very high among children aged five years old [3]. In Thailand, the national oral health survey reported similar findings, indicating that the disease was prevalent (79%) among preschool children [3]. In many places, ECC is mostly left unrestored or untreated, possibly leading to dental infection and toothache. This eventually affects the quality of life and well-being of young children [4]. Following the American Academy of Pediatric Dentistry, ECC is defined as “the presence of one or more decayed (noncavitated or cavitated lesions), missing (due to caries) or filled tooth surfaces in any primary tooth in a child under the age of six” [5]. The etiology of ECC is complex. Fluoride exposure, sugar consumption, and infant feeding practices are significantly associated with ECC [6]. 

Breastfeeding is a vital and natural behavior for the growth and development of infants and young children. The American Academy of Pediatrics recommends that infants should be exclusively breastfed for the first six months, with the continuity of breastfeeding alongside complementary diets for another year or longer [7]. Human breast milk feeding is defined as the practice of an infant only being breastfed or fed human breast milk from a bottle. Breastfeeding is a crucial strategy to reduce infant mortality because it provides essential nutrients for growth and development and helps boost the infant’s immune system [8]. Several studies have shown that breastfeeding reduces the risk of numerous gastrointestinal and respiratory tract infections, atopic eczema, and other allergic disorders [9]. 

At present, breastfeeding has been encouraged and promoted in many countries worldwide [10]. In Thailand, the Ministry of Public Health is aiming to increase the rate of exclusive breastfeeding up to at least 50% by 2025 [11]. A Cochrane review concluded that children with exclusively breastfeeding for six months experienced less morbidity but no risk reduction in dental caries was reported [12]. Following the WHO recommendation, exclusive breastfeeding should continue until two years of age or beyond. ECC prevention should align with international initiatives and be integrated into contemporary primary care systems [13]. 

However, the association between breastfeeding and ECC is currently inconsistent [6,14]. Studies showed that prolonged breastfeeding was associated with ECC [14,15]. Controversially, another study reported no relationship between prolonged breastfeeding and dental caries in young children [16]. Recently, a systematic review concluded that breastfeeding until two years of age did not increase caries risk [6]. The cause of ECC is known to be multifactorial, including biological, behavioral, social, and environmental circumstances [17], and improper feeding patterns may pose an increased risk of developing ECC [18]. 

Most of the studies focused on the duration of breastfeeding and ECC. To date, there is limited information regarding the level of dental plaque and other modifiable risk factors in relation to ECC among exclusively breastfed children with high caries risk. We hypothesized that breastfeeding pattern and oral health-related behaviors may associate with ECC development and severity in this population. Thus, this study aimed to investigate ECC prevalence, caries experience, and the intensity of ECC and its risk factors among 9- to 18-month-old exclusively breastfed children. The results of the present study are beneficial for health care practitioners to provide preventive guidance for parents and caregivers of exclusively breastfed children.

## 2. Materials and Methods 

The Institutional Review Board of Chulalongkorn University (IRB no.: HREC-DCU 2011–004) approved the present study. Written consent was obtained from the parent of each study child. The current study was implemented in full accordance with the World Medical Association Declaration of Helsinki.

The study was conducted at Queen Sirikit National Institute of Child Health, Bangkok, Thailand, from October 2011 to September 2012. Eligibility criteria were 9- to 18-month-old children who were exclusively breastfed from birth to 6 months old and full breastfeeding (breast milk with or without water, other liquids, or food but not formula) until the day of examination and whose mothers were able to write and read in the Thai language. All children who had attended the Well Baby Program in the pediatric clinic for routine vaccination were invited. Exclusion criteria were children who had a major systemic illness. The participating children were examined in the consultation room at the pediatric clinic. 

Regarding the sample size estimation, the ECC prevalence of Thai children aged 12 and 18 months was 22.8% and 66.8% at 18 months, respectively [19]. In the present study, we recruited children aged 9–18 months old, thus, the overall anticipated prevalence would be around 50%. The desired precision of estimation was set as 5%. With the confidence interval set as 95% (alpha = 0.05), 386 children were needed in this study. With the anticipated response rate at 70%, at least 550 dyads needed to be invited.

Mothers of the study children were interviewed using a structured questionnaire regarding the child’s demographic background, breastfeeding behaviors, oral health-related behaviors, dietary practices, and medication intake by an independent interviewer. Before conducting a study, an examiner (P.C.) was trained and calibrated with a specialist in pediatric dentistry (C.T.). The result of the Kappa statistics during the calibration process was 0.9. The study children were examined by a calibrated and trained dentist (P.C.) who was not aware of the mothers’ responses in the interviews. The dental examination was performed in a knee-to-knee position using a dental probe and a dental mirror. Caries was also diagnosed following the WHO diagnostic criteria [20], modified to record the noncavitated lesions or initial lesions according to Warren and colleagues [21]. In the present study, the number of decayed (noncavitated or cavitated), missing due to caries, and filled teeth (d_1_mft) and tooth surfaces (d_1_mfs) were calculated for each participating child. 

Regarding the assessment of dental plaque, four anterior maxillary teeth were observed using the Greene and Vermillion index [22]. A trained examiner used a blunt probe to horizontally scrape the tooth surface at the incisal third, middle third, and cervical third of the anterior maxillary teeth. Then, the amount of plaque on the explorer and the location of plaque accumulation was visually observed. The criteria used in the study were as follows: level 3 = plaque accumulation up to the incisal third; level 2 = plaque accumulation up to the middle third; level 1 = plaque accumulation on the cervical third; level 0 = no presence of dental plaque. The scores of each upper maxillary anterior tooth were summed up and then divided by the total number of teeth examined.

We analyzed the data using the software SPSS 24.0 for Windows (IBM Corp., Armonk, NY, USA). The prevalence of ECC, d_1_mft, d_1_mfs, and intensity of ECC (I-ECC) were calculated. The I-ECC was calculated by dividing the d_1_mft score by the number of erupted teeth. The Shapiro–Wilk test was adopted to test the normality of d_1_mfs score and I-ECC. Because the data were not normally distributed (Shapiro–Wilk test, *p* < 0.05), the Mann–Whitney U test was used to study the distribution of d_1_mfs scores and I-ECC according to the children’s age. Logistic regression was adopted to analyze the relationship of each independent variable and the dependent variable or presence of ECC (yes or no). Negative binomial regression was used to analyze the association between variables and d_1_mfs scores. Poisson regression was used to identify factors that correlated with I-ECC. All potential variables (*p* < 0.05) in the bivariate analysis were inserted as covariates in the multivariate regression model. The backward stepwise procedure was adopted to remove variables that were insignificant (*p* > 0.05) from the model. The final regression model comprised the statistically significant variables. The level of statistical significance in the present study was set at 0.05 for all tests.

## 3. Results

Out of the 560 eligible dyads, 513 children (47% males and 53% females) who had been fed only breast milk for at least six months participated in the study. The children’s mean age was 13.6 months. Two hundred and eighty-four children were 9‒12 months old, whereas 229 children were 13‒18 months old. The prevalence of ECC (including either noncavitated or cavitated lesions) was 42.5% (Table 1). The mean (SD) d_1_mft and d_1_mfs scores were 1.07 (1.41) and 1.34 (1.99), respectively. The mean d_1_mft and d_1_mfs scores and I-ECC according to the children’s age, are shown in Table 2. The older children (13–18 months old) had significantly higher d_1_mft and d_1_mfs scores and I-ECC compared to the younger children (9–12 months old) (Wilcoxon rank-sum test, *p* < 0.001).

Bivariate analysis of potential variables related to the prevalence of ECC, caries experience (d_1_mfs score), and I-ECC are displayed in Table 3. Several significant factors associated with higher ECC prevalence were as follows: higher dental plaque scores, breastfeeding to sleep, ad-lib feeding, no oral cleaning, cleaning less than twice a day, starting oral cleaning after six months old, and not using fluoride toothpaste (logistic regression, *p* < 0.05). Similarly, a higher level of dental plaque, breastfeeding to sleep, no oral cleaning, cleaning less than twice a day, and not using fluoride toothpaste were significantly associated with higher d_1_mfs scores (negative binomial, *p* < 0.05). Three significant variables (a higher plaque score, breastfeeding to sleep, and no oral cleaning) were significantly associated with I-ECC (Poisson regression, *p* < 0.05). 

Table 4 shows the results of the final model of significant factors related to ECC prevalence, caries experience (d_1_mfs), and I-ECC. After adjusting for potential confounding factors, children’s age, breastfeeding to sleep, oral cleaning, and dental plaque were significantly associated with ECC prevalence (*p* < 0.05). Children with dental plaque covering more than middle-third of their tooth surfaces had a significantly higher chance of having ECC, 75.60 times as likely (95% CI: 40.19–142.20, *p* < 0.001), compared to those with less dental plaque. Children who were breastfed to sleep were 2.85 times (95% CI: 1.48–5.49, *p* = 0.002) to develop ECC, compared to those without this behavior. Older children who received no oral cleaning had a higher chance of having ECC (*p* < 0.05). Regarding the risk factors related to caries experience (d_1_mfs), four significant variables were level of dental plaque, oral cleaning practice, breastfeeding to sleep, and children’s age (*p* < 0.05), whereas the level of dental plaque was the only significant variable associated with I-ECC (*p* < 0.001).

## 4. Discussion

ECC was reported to be very high in Southeast Asia [3]. So far, risk factors of ECC in exclusively breastfed children have not been well documented in developing countries in this region, compared with those in developed countries, where living conditions and child-rearing practices are considerably different. Up to now, few studies have been conducted on exclusively breastfed children in Thailand. Among children with various feeding practices, a previous study reported that ECC prevalence was very high in Thailand: 2% in 9-month-olds, 22.8% in 12-month-olds, and 68.1% in 18-month-olds [19]. Similarly, our study found that the ECC prevalence of breastfed children was high (42.5%) and escalated with age. Several behavioral risk factors, such as oral cleaning and feeding practices, were found to be significant. 

Among all potential risk factors studied, dental plaque accumulation was the key risk factor for ECC with regard to the prevalence, magnitude, and intensity. Children who had dental plaque accumulation covering over middle-third of the total anterior tooth area had a significantly higher risk of developing ECC (75 times) than those who had less dental plaque. These findings are in agreement with the care pathways for managing caries in young children [23]. An individualized caries risk assessment by collecting information regarding the amount of dental plaque, dental hygiene, and feeding practices should be developed based on existing programs such as vaccination programs to prevent ECC in these high caries risk children. Our results showed that children who did not have their teeth cleaned had a significantly higher caries risk (8.5 times), compared with those who did. This finding is in line with previous studies supporting the importance of cleaning the erupting teeth and soft tissues of infants [24,25], possibly allowing children to become accustomed to oral cleaning practices.

Interestingly, although the recent evidence demonstrated the effectiveness of fluoride toothpaste in decreasing the caries increment in primary teeth [26], the present study showed no correlation between ECC and the use of fluorides in the very young age population. The reason might be that most of the previous studies were done in the older age group than the population we studied. The effectiveness of fluoride toothpaste in infants needs to be further studied. Another systematic review also demonstrated that the frequency of tooth brushing was associated with caries incidence or increment [27]. On the contrary, the frequency of daily oral cleaning (once or less vs. twice or more) was not associated with caries status in the study population. These conflicting results may be due to a unique characteristic of caries development in very young children. Possibly, the reported higher frequency of oral hygiene practices may not truly reflect the higher efficiency of plaque removal. The quality of oral cleaning, as assessed by observing dental plaque, would be a more valid risk indicator of ECC. These findings concur with a previous study indicating that the self-reported frequency and method of oral cleaning practices by caregivers did not determine the cleanliness nor did it correlate with caries development [28]. 

Inappropriate nursing behavior is another risk factor for dental caries development. Several studies have demonstrated that a child falling asleep while suckling milk with various feeding practices had an increased risk of ECC development [15,29]. These findings are in accordance with the present study. The study children who were breastfed until asleep were at a higher risk of caries than those who were not. During the night, the salivary flow rate typically slows down, thus reducing the ability of flushing milk residue, and eventually facilitating caries initiation. Based on the results of the present study, the higher plaque score posed the greatest risk among all factors studied (Table 4). This implies that dental plaque plays a major role, whereas the reported oral health-related behaviors may be a secondary concern. 

Notably, ECC prevalence increased markedly in older children by month. Screening and identifying children, in particular those with high caries risk, at a very young age may be crucial to prevent new caries and reduce the intensity of the disease. Our findings showed that most of the lesions in infants (9–12 months old) were categorized as noncavitated caries, which can be reversed or halted by brushing with fluoride toothpaste and modifying feeding practices. Minimally invasive interventions such as sodium fluoride varnish and silver diamine fluoride should be adopted in controlling ECC at the early stage [30]. However, in Thailand and other places, parents are unlikely to bring their baby or young child to see a dentist. Primary health care providers may play a vital role to promote infants’ and toddlers’ oral health [31]. They can perform dental screenings by merely lifting a child’s upper lip to determine the presence of dental plaque on upper anterior teeth and assess the caries risk of children during the first year of life. Nevertheless, further study is required to assess the benefit of routine oral screenings and noninvasive interventions, if necessary, performed by primary health care providers.

The present study had some strengths, such as a sufficient sample size. Information on potential caries risk factors, including feeding patterns and oral hygiene practices, was comprehensively collected for controlling the confounding factors. Recall bias regarding the feeding and oral hygiene practices at the time that children were 9–18 months old would be lower, compared to that of the ECC study conducted in children aged 3–5 years. However, the study had some inherent limitations. The duplicated examinations for assessing the reliability of examiner were impractical in this study setting. Some potential confounders that may affect the outcomes such as family income, parent’s educational level, and Streptococcus mutans and lactobacilli were not included in the analysis. Based on the nonprobability sampling used, sampling bias could occur. Caution is warranted in interpreting these results to make inferences about general populations. The nature of the cross-sectional study that evaluated the ECC prevalence at one point in time may hinder the causal relationship between ECC and child-rearing practices. 

A well-designed cohort study adopting the probability sampling method is required to confirm the effect of feeding habits and oral cleaning behaviors on ECC among exclusively breastfed children from birth to preschool age. Despite the unfavorable oral health outcomes reported in the present study, the benefits of breastfeeding are unparalleled. We affirm that breastfeeding should not be discouraged. Instead, key determinants of ECC among breastfed children should be further investigated, and effective interdisciplinary-based preventive measures should be implemented in the early childhood stage.

## 5. Conclusions

The prevalence of ECC is high among exclusively breastfed children aged 9–18 months in Thailand. Their ECC prevalence and caries experience are significantly associated with the level of dental plaque, children’s age, breastfeeding to sleep, and oral cleaning practices, whereas the I-ECC is only associated with the level of dental plaque.

## Figures and Tables

**Table 1 ijerph-17-03194-t001:** Prevalence of early childhood caries (ECC) according to the age of the children.

Age (Months)	No. of Children	No. of Children (%)		Total no. of Children with ECC *** (%)
		Noncavitated Lesions *	Cavitated Lesions **	
9–12	284	80 (28.2%)	1 (0.4%)	81 (28.5%)
13–18	229	94 (41.0%)	43 (18.8%)	137 (59.8%)
Total	513	174 (33.9%)	44 (8.6%)	218 (42.5%)

* Children who had noncavitated carious lesion(s) without cavitated lesions. ** Children who had at least one cavitated carious lesion. *** Children who had ECC (either noncavitated or cavitated carious lesions).

**Table 2 ijerph-17-03194-t002:** The mean of d_1_mft, d_1_mfs, and I-ECC indices according to the age (month).

Age (Months)	d_1_mft (Mean ± SD)	d_1_mfs (Mean ± SD)	I-ECC (Mean ± SD)
9–12	0.52 ± 0.88 ^a^	0.59 ± 1.11 ^b^	0.09 ± 0.16 ^c^
13–18	1.76 ± 1.64 ^a^	2.28 ± 2.40 ^b^	0.15 ± 0.14 ^c^
Total	1.07 ± 1.41	1.34 ± 1.99	0.11 ± 0.16

Wilcoxon rank-sum (Mann–Whitney) test. Same letter indicates statistically significant difference (*p*-value < 0.001).

**Table 3 ijerph-17-03194-t003:** Bivariate analysis of potential variables related to the prevalence of ECC, the mean of d_1_mfs score, and I-ECC.

Variables	*n*	Caries Status (Dependent Variables)								
		ECC			d_1_mfs			I-ECC		
		Prevalence (95% CI)	Unadjusted OR (95% CI)	*p*	Mean (95% CI)	IRR (95% CI)	*p*	Mean (95% CI)	IRR (95% CI)	*p*
Gender										
Male	240	43% (36%–49%)	0.97 (0.68–1.37)		1.45 (1.19–1.71)	0.83 (0.63–1.17)		0.13 (0.10–0.14)	0.91 (0.55–1.50)	
Female	273	42% (36%–48%)	1	0.856	1.24 (1.19–1.02)	1	0.332	0.11 (0.09–0.13)	1	0.716
Frequency of feeding during day time										
≤4	96	42% (33%–52%)	1		1.41 (0.5–1.86)	1		0.13 (0.10–0.17)	1	
>4	417	42% (38%–47%)	0.99 (0.58–2.29)	0.693	1.33 (1.14–1.51)	0.94 (0.57–1.92)	0.769	0.11 (0.10–0.13)	0.86 (0.37–2.69)	0.618
Frequency of feeding at night										
≤3	337	41% (35%–46%)	1		1.25 (1.04–1.46)	1		0.11 (0.10–0.13)	1	
>3	176	46% (39%–53%)	1.24 (0.86–1.80)	0.243	1.5 (1.21–1.80)	1.19 (0.87–1.65)	0.266	0.13 (0.10–0.15)	1.08 (0.46–1.83)	0.756
Feeding duration during day time										
≤10 min	480	43% (38%–47%)	1		1.32 (1.15–1.50)	1		0.12 (0.10–0.13)	1	
>10 min	33	42% (26%–59%)	0.99 (0.49–2.03)	0.993	1.64 (0.77–2.50)	1.23 (0.67–2.29)	0.493	0.12 (0.06–0.18)	1.04 (0.38–2.84)	0.94
Feeding duration at night										
≤10 min	388	43% (38%–48%)	1		1.36 (1.16–1.56)	1		0.12 (0.11–0.14)	1	
>10 min	125	40% (17%–29%)	0.87 (0.58–1.32)	0.517	1.28 (0.94–1.62)	0.94 (0.66–1.35)	0.738	0.11 (0.08–0.14)	0.92 (0.50–1.68)	0.79
Breastfeeding to sleep										
Yes	319	54% (49%–60%)	3.9 (2.61–5.81)		1.76 (1.52–2.00)	2.7 (1.96–3.71)		0.15 (0.14–0.17)	2.5 (1.32–4.70)	
No	194	23% (17%–29%)	1	<0.001	0.65 (0.45–0.84)	1	<0.001	0.06 (0.04–0.08)	1	0.005
Ad-lib feeding										
Yes	401	45% (41%–50%)	1.75 (1.13–2.73)		1.06 (0.71–1.41)	1.33 (0.91–1.95)		0.09 (0.07–0.12)	1.35 (0.70–2.64)	
No	112	32% (23%–41%)	1	0.013	1.42 (1.22–1.62)	1	0.134	0.13 (011–0.14)	1	0.373
Water is given after breast milk										
Yes	122	47% (38%–56%)	1.25 (0.83–1.89)		1.68 (1.27–2.11)	1.37 (0.96–1.94)		0.14 (0.11–0.17)	1.29 (0.74–2.24)	
No	391	44% (37%–52%)	1	0.28	1.23 (1.04–1.42)	1	0.08	0.11 (0.10–0.13)	1	0.373
Frequency of between-meals at day time										
≤3	163	44% (37%–52%)	1		1.31 (1.02–1.61)	1		0.13 (0.10–0.15)	1	
>3	350	42% (37%–47%)	0.9 (0.62–1.31)	0.6	1.35 (1.14–1.57)	1.03 (0.74–1.43)	0.854	0.11 (0.10–0.13)	0.89 (0.53–1.52)	0.676
Frequency of between-meals at night time										
≤1	312	44% (38%–49%)	1		1.34 (1.12–1.56)	1		0.12 (0.10–0.14)	1	
>1	201	40% (34%–47%)	0.86 (0.60–1.24)	0.419	1.34 (1.06–1.62)	1 (0.72–1.36)	0.983	0.12 (0.09–0.14)	1 (0.60–1.67)	0.999
Oral cleaning										
Yes	469	38% (33%–42%)	1		1.03 (0.88–1.17)	1		0.1 (0.09–0.12)	1	
Never	44	93% (86%–100%)	25 (6.66–100)	<0.001	4.7 (3.89–5.52)	4.35 (2.94–7.14)	<0.001	0.29 (0.26–0.33)	2.94 (1.56- 5.56)	0.001
Starting age of cleaning practice										
≤6 months	498	42% (37%–46%)	1		1.31(1.14–1.48)	1		0.12 (0.10–0.13)	1	
>6 months	15	73% (51%–96%)	3.86 (1.21–12.31)	0.022	2.33 (1.19–3.47)	1.78 (0.75–4.21)	0.19	0.25 (0.15–0.35)	2.14 (0.75–6.74)	0.155
Frequency of daily oral cleaning										
<2	193	52% (45%–59%)	1		1.75 (1.44–2.06)	1		0.15 (0.13–0.17)	1	
≥2	320	37% (32%–42%)	0.54 (0.38–0.78)	0.001	1.09 (0.89–1.29)	0.62 (0.46–0.85)	0.003	0.1 (0.08–0.12)	0.67 (0.40–1.12)	0.125
Ever visited a dentist?										
Never	351	41% (36%–46%)	1		1.28 (1.07–1.48)	1		0.11 (0.10–0.13)	1	
Yes	162	46% (37%–54%)	0.8 (0.54–1.16)	0.237	1.48 (1.17–1.79)	0.86 (0.61–1.19)	0.373	0.13 (0.10–0.15)	0.9 (0.53–1.52)	0.688
Child received fluoride supplement?										
No	351	42% (37%–47%)	1		1.33 (1.12–1.54)	1		0.11 (0.10.0.13)	1	
Yes	162	43% (36%–51%)	0.92 (0.63–1.34)	0.678	1.36 (1.06–1.67)	0.98 (0.70–1.35)	0.882	0.13 (0.10–0.15)	0.9 (0.53–1.55)	0.728
Child using fluoride toothpaste?										
No	358	40% (35%–45%)	1		1.17 (0.98–1.34)	1		0.11 (0.09–0.12)	1	
Yes	155	49% (41%–56%)	0.68 (0.47–0.99)	0.049	1.74 (1.36–2.13)	0.67 (0.48–0.92)	0.016	0.14 (0.11–0.17)	0.75 (0.47–1.26)	0.27
Plaque index										
<2	288	7% (4%–10%)	1		0.16 (0.09–0.24)	1		0.02 (0.01–0.02)	1	
≥2	225	88% (84%–92%)	98.3 (53.6–180.3)	<0.001	2.85 (2.57–3.13)	17.5 (12.3–25.4)	<0.001	0.25 (0.23–0.27)	15.4 (6.0–39.66)	<0.001

Plaque index = modified Greene and Vermillion index; OR = Odds ratio; IRR = Incidence rate ratio; 95% CI = 95% Confident interval.

**Table 4 ijerph-17-03194-t004:** Multivariate regression models for caries prevalence, d_1_mfs, and I-ECC.

Multivariate Logistic Regression Model for Caries Prevalence			
Risk Factors	Adjusted OR	95% CI	*p*-Value
Age (month)	1.10	1.00–1.20	0.048
Breastfeeding to sleep			
No	1		0.002
Yes	2.85	1.48–5.49	
Oral cleaning			0.014
Yes	1		
Never	8.51	1.53–47.14	
Plaque index			<0.001
<2	1		
≥2	75.60	40.19–142.20	
**Multivariate Negative Binomial Regression Model for d_1_mfs**			
Risk Factor	IRR	95% CI	*p*-value
Age (month)	1.08	1.04–1.13	<0.001
Breastfeeding to sleep			
No	1		0.001
Yes	1.58	1.14–2.17	
Oral cleaning			0.001
Yes	1		
Never	1.92	1.30–2.84	
Plaque index			<0.001
<2	1		
≥2	11.70	8.17–16.76	
**Multivariate Poisson Regression Model for I-ECC**			
Risk Factor	IRR	95% CI	*p*-Value
Plaque index			
<2	1		
≥2	14.82	10.46–20.99	<0.001

Plaque index = modified Greene and Vermillion index; OR = Odds ratio; IRR = Incidence rate ratio; 95% CI = 95% confident interval.

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
