# Peer review of "Early Childhood Caries and Its Associated Factors among 9- to 18-Month Old Exclusively Breastfed Children in Thailand: A Cross-Sectional Study"

_ijerph, 2020, doi:10.3390/ijerph17093194_

Round 1
Reviewer 1 Report
This manuscript describes an interesting study of the impact of breast feeding on early childhood caries and the associated factors in a large sample of subjects from Thailand. It adds important evidence to the field and helps to identify key preventive messages that may be directed to mothers. I presume that the information on fluoride supplementation and toothpaste use refers to the infants, but perhaps this could be clarified. Although the fluoride variables were not statistically significantly correlated with ECC, a sentence (with relevant reference citations) might be added to nevertheless highlight that evidence exists for the value of fluoride intervention in caries development.
Author Response
Ijerph-770150
Reviewer 1
This manuscript describes an interesting study of the impact of breast feeding on early childhood caries and the associated factors in a large sample of subjects from Thailand. It adds important evidence to the field and helps to identify key preventive messages that may be directed to mothers. I presume that the information on fluoride supplementation and toothpaste use refers to the infants, but perhaps this could be clarified. Although the fluoride variables were not statistically significantly correlated with ECC, a sentence (with relevant reference citations) might be added to nevertheless highlight that evidence exists for the value of fluoride intervention in caries development.
Thanks for your comments and suggestion
This point is added in Page 6 (table 3) and Discussion section Page 9 line 23-27.
Reviewer 2 Report
The paper shows that some risk factors were associated with ECC among 9- to 18-month-old exclusively breastfed children in Thailand in a cross-sectional study.
This is an interesting study. However, I would like to make some points regarding the manuscript. The article needs to be revised. Especially, the authors should follow the STROBE guideline.
INTRODUCTION
1) Please set one topic per one paragraph. Thus, the authors should revise each paragraph carefully.
2) There are many reviews and papers in this field including Cochrane Database Syst Rev. What are the unclear things in this research field and new things in this study? Please define them clearly in the text.
3) Please add the hypothesis before the aim.
MATERIALS AND METHODS
1) Please add some parts following the STROBE guideline, such as sample size estimation, flowchart, experimental period (one year or a few months), bias, etc.
2) Who did investigate the oral conditions? How about the calibration and its results?
RESULTS
1) What is “adjusted OR” in the Table 3?
2) Please add the number in addition to percentage of each category in the Table 3.
3) Some 95CI% ranges are broad. Please check the number of each case. If some are small, please re-analyze the data and change the logistic regression model.
DISCUSSION
1) Please set one topic per one paragraph. Thus, the authors should revise each paragraph carefully.
2) Please add more comments about limitation, such as small number of participants, sampling bias, no generalizability, no data of confounders, etc.
3) The authors can’t state “increase” in a cross-sectional study. Please revise the conclusion.
Author Response
"Please see the attachment."

Round 2
Reviewer 1 Report
The authors have adequately addressed the points I raised
Author Response
Thank you very much for your kind consideration.
Reviewer 2 Report
The paper was overall improved. However, there are some issues. The paper should be revised.
INTRODUCTION
1) The topic sentence of third paragraph is “At present, breastfeeding has been encouraged and promoted in many countries worldwide [10].” However, the authors include the relationship between ECC and breastfeeding. Please set one topic per one paragraph. Thus, the authors should revise the paragraph carefully.
MATERIALS AND METHODS
1) Please add some references and alpha value and power level in the sample size estimation. Why did the authors set 50%? Please add appropriate the references.
RESULTS
1) I understand the status. Thus, please delete the appendix.
2) My last questions are below.
“Some 95CI% ranges are broad. Please check the number of each case. If some are small, please re-analyze the data and change the logistic regression model.”
For example, The number of participants with Plaque index ≤1 was 220 and in these group, only 3 % (7 persons) had ECC. The number was quite small and then, the 95%CI was very broad or “5.83-142.60”. In this case, the model is not appropriate. The authors should cancel the model.
The authors should change other data, too. Please re-analyze the data carefully.
DISCUSSION
1) My last questions are below.
“The authors can’t state “increase” in a cross-sectional study. Please revise the conclusion.”
The authors did not revise it, Their ECC prevalence “increases” significantly with a high level of dental plaque, children’s age, ad-lib 86 feeding, breastfeeding to sleep, no oral cleaning, and delayed oral cleaning practices.
Furthermore, the logistic regression analyses should be changed, then the conclusion will be changed.
Round 3
Reviewer 2 Report
The revision is acceptable.